# Alteration in sB7-H4 Serum Levels and Placental Biomarker Expression after Therapeutic Plasma Exchange in Early-Onset Preeclampsia Patients

**DOI:** 10.3390/ijms252011082

**Published:** 2024-10-15

**Authors:** Liyan Duan, Yuyang Ma, Beatrix Reisch, Elina Hadrovic, Pawel Mach, Rainer Kimmig, Michael Jahn, Angela Köninger, Antonella Iannaccone, Alexandra Gellhaus

**Affiliations:** 1Department of Gynecology and Obstetrics, University of Duisburg-Essen, 45147 Essen, Germany; duanliyan1988@126.com (L.D.); mayuyang0921@gmail.com (Y.M.); beatrix.reisch@uk-essen.de (B.R.); elina.hadrovic@uk-essen.de (E.H.); pawel.mach@uk-essen.de (P.M.); rainer.kimmig@uk-essen.de (R.K.); angela.koeninger@barmherzige-regensburg.de (A.K.); 2Department of Nephrology, University of Duisburg-Essen, University Hospital Essen, 45147 Essen, Germany; michael.jahn@krupp-krankenhaus.de; 3Department of Gynecology and Obstetrics, University Clinic St. Hedwig of the Order of St. John, 93049 Regensburg, Germany

**Keywords:** therapeutic plasma exchange, preeclampsia, B7-H4, soluble fms-like tyrosine kinase 1, soluble Endoglin, preeclampsia therapy, placental dysfunction, preterm birth

## Abstract

Therapeutic plasma exchange (TPE) is a widely used treatment for numerous diseases including pregnancy-related conditions. Our prior study on 20 early-onset preeclampsia patients undergoing TPE revealed a significant extension in pregnancy duration and reduced serum levels of sFlt-1, sFlt-1/PlGF, and sEndoglin. Here, we investigated the impact of TPE on serum sB7-H4, an immunological checkpoint molecule, and placental proteins (Flt-1, Eng, B7-H4, iNOS, TNF-α) in TPE-treated early-onset preeclampsia patients (N = 12, 23 + 2–28 + 5 weeks), conventionally treated counterparts (N = 12, 23 + 5–30 weeks), and gestational age-matched controls (N = 8, 22 + 4–31 + 6 weeks). Immunoblotting, ELISA, and co-immunohistochemistry were used for biomarker analysis, including placental inflammation factors (iNOS, TNF-α). The results showed that TPE extended pregnancy by a median of 6.5 days in this cohort of early-onset preeclampsia. Serum sB7-H4, sFlt-1, and sEndoglin levels decreased, along with reduced expression of their membrane-bound proteins in placental tissue upon TPE treatment. Moreover, TPE-treated patients displayed reduced placental inflammation compared to preeclampsia patients receiving standard-of-care treatment. In conclusion, TPE may improve pregnancy outcomes in early-onset preeclampsia by lowering circulating levels of sB7-H4, sFlt-1, and sEndoglin, as well as reducing placental inflammation. This translational approach holds promise for enhancing placental function and extending gestation in high-risk pregnancies including very preterm PE or HELLP cases.

## 1. Introduction

Preeclampsia (PE), affecting 2–8% of pregnancies and directly contributing to 10–15% of maternal fatalities worldwide [1], is one of the most serious obstetrical complications [2]. In selected cases, it can rapidly evolve into HELLP syndrome (hemolysis, elevated liver enzymes, and low platelets). Annually, approximately 4 million women receive a diagnosis of PE, leading to the mortality of over 70,000 women and 500,000 infants [3]. The high-risk factors for PE include but are not limited to previous PE pregnancy, advanced age, high body mass index (BMI), and paternal factors [4,5]. Despite the gravity of PE, there exists no therapeutic intervention apart from the delivery of the fetus. Immediate delivery may be warranted when there is a change in maternal or fetal circumstances in severe PE or after reaching 34 weeks of gestation [6]. However, the management of early-onset PE poses a significant challenge. Early-onset PE (occurring before 34 weeks of gestation) is linked to higher fetal morbidity and mortality than late-onset disease (presenting at or after 34 weeks) [7]. The mother may gain a benefit from delivery, but the preterm delivery is a risk for neonatal well-being because of several severe short- and long-term sequelae of immaturity. Expectant therapy in women with early-onset PE before 34 weeks’ gestation may minimize newborn problems and the length of stay in a newborn intensive care unit (NICU), but may exacerbate the maternal disease [8,9,10]. Therefore, in severe early-onset PE, especially at the gestational threshold of newborn viability (22–24 weeks of gestation), the current challenge is to prolong the gestational period as much as possible without threatening the mother’s health.

A combination of abnormal placentation and ischemia could lead to the complex process of PE development [11,12]. This combination causes the release of pro-inflammatory and anti-angiogenic proteins into the maternal circulation, which ultimately causes endothelial dysfunction and the clinical syndrome seen in PE patients [13].

Soluble fms-like tyrosine kinase-1 (sFlt-1) and placenta growth factor (PlGF) represent two extensively investigated and predictive biomarkers, particularly concerning the early-onset of PE [14]. sFlt-1, an anti-angiogenic factor that inhibits neovascularization, exhibits elevated levels in individuals with PE [15,16]. sFlt-1 reduces the amount of effective vascular endothelial growth factor (VEGF) by binding to it. Its excessive expression in PE can result in insufficient angiogenesis [17]. The level of sFlt-1 in the bloodstream is correlated with the severity of PE and the proximity to the development of hypertension or proteinuria. Concurrently, PlGF, which plays a crucial role in the development of blood vessels during pregnancy, is diminished, resulting in an elevated sFlt-1 to PlGF ratio in patients with PE [16,18]. Previously, the sFlt-1/PlGF ratio has been employed to formulate PE management strategies [1]. This ratio is utilized to classify individuals as either at extremely low risk of developing PE within four weeks (with a ratio under 38), at intermediate risk (with ratio 38–85 for early-onset disease or 38–110 for late-onset illness), or at an elevated probability of experiencing PE (with a ratio exceeding 85 or 110).

Endoglin (Eng), a coreceptor for transforming growth factor beta 1 (TGFβ-1) and transforming growth factor beta 3 (TGFβ-3), is abundantly present on the cell membranes of vascular endothelium and syncytiotrophoblasts (STBs) in the placenta during pregnancy [19]. PE induces an upregulation of placental Eng, which can further undergo modification into soluble Endoglin (sEng) and be released into the maternal bloodstream [19]. sEng is a protein with the potential to suppress TGFβ-1 signaling in the vasculature [19,20]. Adenoviral-mediated sFlt-1 and sEng overexpression in rats resulted in severe vascular damage, nephrotic-range proteinuria, severe hypertension, a condition akin to the HELLP syndrome (hemolysis, increased liver enzymes, and low platelets), and fetal growth restriction (FGR) [19]. Thus, sEng and sFlt-1, two anti-angiogenic proteins affecting VEGF and TGFβ signaling pathways, respectively, may interact to cause endothelial dysfunction and severe PE [21]. Moreover, PE is thought to be associated with a failure to develop proper immunological tolerance, characterized by an excess of Th1-type responses [22,23]. B7-H4 has been found to be important in T-cell inhibition [24,25]. In a preceding retrospective cohort study by us, we observed significantly elevated levels of soluble B7-H4 (sB7-H4) in maternal blood within PE patients in comparison to the controls [26]. Furthermore, the levels of B7-H4 in placental villous tissue were significantly higher in women with early-onset PE compared to healthy controls [27].

Therefore, sFlt-1/PlGF, sB7-H4, and sEng play an important role in the pathogenesis of PE.

Therapeutic plasma exchange (TPE) is an extracorporeal therapy that focuses on replace plasma, thus removing soluble and pathogenic components from the circulation contained in it. It is the apheresis method most widely used throughout the years [28]. A recent study by us demonstrated that TPE treatment can significantly attenuate the serum sFlt-1, sFlt-1/PlGF and sEng levels in early-onset PE [29]. Furthermore, this study, including 20 patients with PE undergoing TPE compared to 21 PE patients with standard-of-care treatment but without TPE intervention, revealed a significant extension in gestational duration with TPE treatment. On average, in pregnancies characterized by early-onset PE and subjected to TPE, the median duration was prolonged by 6 days from the initiation of treatment. In contrast, the standard treatment led to a median increase in the time from hospitalization to delivery of only 1 day [29].

In this study, we will further investigate the impact of TPE on serum sB7-H4 levels in patients with early-onset PE. Since the primary sources of serum sFlt-1 [16], sEng [19], and sB7-H4 [27,30] in PE patients are membrane proteins in the corresponding placental tissues, which undergo possible alternative splicing and release during TPE, the protein levels of Flt-1, Eng, and B7-H4 in the placenta will additionally be assessed. Moreover, inflammatory markers in placental tissues will be explored to gain insights into functional alterations within the placenta.

## 2. Results

### 2.1. Patient’s Characteristics

Table 1 provides an overview of the clinical information for three patient groups: the control group (N = 8), the PE group treated with standard of care (abbreviated as PE group, N = 12), and the PE patients who underwent TPE treatment (abbreviated as PE + TPE group, N = 12). There were no significant differences in maternal age, gestational age, pregnancy BMI before birth, alanine aminotransferase (ALT) levels, lactate dehydrogenase (LDH) levels, and neonatal 5-min Apgar score among the three groups (Table 1). Systolic and diastolic blood pressures were elevated, and platelet counts were reduced in both the PE and PE + TPE patient groups, relative to the control group, aligning with the diagnostic criteria for PE patients (Table 1). Furthermore, systolic and diastolic blood pressures, sFlt-1, and the sFlt-1/PlGF ratio (Appendix A) at admission also did not differ statistically significantly between the PE and PE + TPE group. The newborns in the PE + TPE group had a significantly lower birth weight compared to the control group. The birth weight percentiles in the PE and PE + TPE groups were significantly lower compared to the control group.

Notably, the PE + TPE group displayed statistically significant reductions in AST levels after TPE compared to the PE group with standard-of-care treatment, as previously demonstrated in Iannaccone et al., 2023. In comparison to the preterm labor control group, both PE and PE + TPE patients exhibited statistically significant elevations in the serum levels of sFlt-1 and sFlt-1/PlGF at delivery. Previous studies allow us to conclude that the serum levels of sEng [31] and sB7-H4 [26] in PE patients are higher than those in controls. Importantly, although not statistically significant, the serum levels of these molecules (sFlt-1, sFlt-1/PlGF, sEng, sB7-H4) demonstrated a decreasing trend in the PE + TPE patients as compared to the PE patients who received standard-of-care treatment (Table 1).

### 2.2. Prolongation of Pregnancy and Neonatal Outcome of PE Patients with TPE Treatment

In this patient cohort, among the 12 patients with PE who underwent TPE, the mean frequency of treatment per patient was 5 ± 3.045 times (range: 1–11 times). Among these patients, although not statistically significant, the duration of pregnancy was extended by a median of 6.5 days (range: 1–23 days). In contrast, PE patients who did not undergo TPE showed an average pregnancy extension of only 3.5 days (range: 1–22 days) after hospitalization (Table 1). All patients in the PE and PE + TPE were delivered by cesarean section; in the control group, the cesarean section rate was 75%. The neonatal survival rates of the control, PE, and PE + TPE groups were 75%, 83.33%, and 58.33%, respectively (Table 1). In the TPE treatment group, 5 out of the 12 patients gave birth at gestational weeks greater than 24 + 0 weeks (24 + 6–28 + 4 w) and no cases of fetal and neonatal death were registered in this group. In total, 7 out of the 12 patients gave birth at gestational weeks less than 24 weeks (23–24 w). Due to the lower gestational age, only two newborns survived the neonatal period.

### 2.3. Changes in the Immunological Marker sB7-H4 in Maternal Serum Associated with Severe PE after TPE Treatment

Our recent study revealed that TPE can significantly attenuate the serum sFlt-1, sFlt-1/PlGF, and sEng levels in PE patients (N of treatments = 95) [29]. In this study, 12 PE patients underwent a total of 60 sessions of TPE. After TPE treatment, the serum levels of sFlt-1, sFlt-1/PlGF, and sEng showed significant reductions compared to those before TPE (Table 2, Figure 1A–C). Additionally, there was an overall decreasing trend in the serum level of sB7-H4. While there was no significant change in median sB7-H4, the mean difference was 29.4% and therefore comparable to the changes in the other examined anti-angiogenic molecules such as sFlt-1 and sEng (Table 2, Figure 1D).

In addition, the first TPE treatment appears to have only a slight, rather than significant, effect in reducing sFlt-1, sFlt-1/PlGF, and sEng levels in patients with PE (Figure 2A–C). sB7-H4 even showed a slight increase after the first TPE treatment (Figure 2D).

### 2.4. Protein Expression of Flt-1, Eng, and B7-H4 in PE Placentas Following TPE Treatment

In comparison to preterm controls and PE patients treated with the standard of care, the placental tissue levels of Flt-1 and B7-H4 were significantly decreased in PE patients treated with TPE (Figure 3A,B,D). Eng protein expression was higher in the PE group than in the control group, whereas there was no difference between patients with PE treated with TPE and those in the control group (Figure 3A,C).

Immunofluorescence analysis showed distinct localization patterns for Flt-1, Eng, and B7-H4 in placental villous tissue. In the PE group without TPE treatment, Flt-1 was strongly expressed in the syncytiotrophoblast (STB), stroma, and fetal blood vessels (Figure 4A–C).

Although the overall placental Eng protein levels did not significantly differ between PE patients treated with TPE and those without detection by immunoblotting (Figure 3C), immunofluorescence staining revealed a more pronounced localization of Eng on both the apical and basal surfaces of the STB in PE patients without TPE treatment. In contrast, the control group and PE patients treated with TPE displayed reduced Eng staining, particularly along the basal side of the STB (Figure 4D–F).

Furthermore, B7-H4 expression appeared significantly stronger in the PE group without TPE treatment compared to both the control group and the TPE-treated PE group (Figure 4G–I).

### 2.5. Placental Inflammation and Apoptosis Levels in PE Patients Following TPE Treatment

Inducible nitric oxide synthase (iNOS) is an enzyme involved in the metabolism of reactive oxygen and nitrogen species, capable of inducing substantial synthesis of nitric oxide (NO). Inflammation can induce iNOS expression [32]. iNOS is only marginally expressed in human placenta under normal conditions. However, the expression of iNOS is increased in the placental tissues of patients with gestational diabetes, preeclampsia, and other inflammation-related complications [32,33].

Our recent study revealed that TPE significantly reduced apoptosis (decreased cleaved caspase 3) and induced proliferation capacity upon increased Cyclin D1 and decreased p21 protein expression in human placental tissue [34]. In this study, our results revealed that the placental tissue levels of iNOS and TNF-α were significantly higher in PE patients without TPE compared to preterm controls and PE patients treated with TPE (Figure 5A–D).

## 3. Discussion

TPE has gained increasing interest in managing pregnancy-related disorders by eliminating soluble substances from the plasma [35]. Its application extends to a range of conditions, including but not limiting to pregnancy-related autoimmune disorders and PE [36]. Numerous studies have shown that as long as the maternal blood volume is maintained during TPE treatment, there are no adverse effects on the fetus [37,38,39]. However, its use during pregnancy initially carried significant risks, such as allergic reactions, hypotension, and infection in mothers [40]. Nonetheless, as expertise has accumulated and technology advanced, the likelihood of these risks occurring has been significantly reduced.

PE at an extremely premature gestational age presents a significant clinical challenge. The only certain cure is delivery, which poses considerable risks of infant death and severe long-term damage owing to preterm birth [41]. Gestational duration continues to be a significant factor in perinatal mortality (stillbirth or neonatal death within 7 days post-partum), despite high standards in perinatal intensive care. The survival rates in infants who were born at 23 and 24 weeks of gestation were 20.0% and 77.8%, respectively [42]. Therefore, it is crucial to extend the gestational week beyond 24 weeks.

TPE has been demonstrated, both by us and others, to effectively prolong gestational age in early-onset PE patients [29,35].

In this study, we performed a retrospective study of a patient cohort of 12 PE patients treated with TPE where we investigated the placenta tissue in addition to the maternal serum and compared the results with placenta tissue from PE patients receiving standard-of-care treatment as well as with preterm controls. Our findings align with previous conclusions, indicating that TPE (with an average frequency of four times per patient) extended the gestational duration by a median of 6.5 days, which is a notable benefit for neonatal outcomes. In our study findings, the neonatal birth weight and survival rates in the PE + TPE group were notably lower compared to the other two groups, likely attributed to the comparatively shorter gestational age observed in the PE + TPE group compared to the control and PE groups. TPE was performed in an experimental setting as a therapeutical approach in cases of extremely early PE with no other options to prolong pregnancy, which is why the gestational age in these cases is lower than in PE cases without TPE.

Furthermore, our results indicate that while an initial unique TPE treatment can reduce harmful circulating factors to some extent, this reduction does not reach statistical significance. Only in cases of repeated TPE treatments was a significant decrease in the serum levels of sFlt-1, sFlt-1/PLGF, sEng, and sB7-H4 detected. Therefore, we hypothesize that repeating TPE therapy during pregnancy may have a greater therapeutic potential for patients with very early-onset PE. It is important to underline that TPE was offered during a historical timeframe with nearly no experience in TPE in very early PE worldwide.

As previously mentioned, the pathology in patients with PE involves impaired placental development and functionality. One manifestation is the elevated serum levels of anti-angiogenic factors, such as sFlt-1, the sFlt-1/PlGF ratio, and sEng, leading to systemic endothelial dysfunction [43]. Endothelial impairment results in increased platelet adhesion, reduced fibrinolysis, and activation of the coagulation cascade [44]. Chaudhry et al. demonstrated that TPE mitigates the circulating platelet aggregates and pro-coagulant substances generated by activated platelets and endothelial cells [45]. Consistent with our prior findings, we observed that TPE effectively reduces the levels of sFlt-1 and the sFlt-1/PlGF ratio. There was also a trend of decreased levels of sB7-H4 following TPE treatment, although statistical significance was not achieved. The immunofluorescence staining showed that B7-H4 was more intensely localized in the STB in early-onset PE patients receiving standard-of-care treatment, indicating a potential correlation between increased B7-H4 expression and the severity of PE pathology in the absence of TPE.

B7-H4 functions as a co-inhibitory molecule on antigen-presenting cells, engaging with an unidentified receptor on T cells to convey negative signaling during T-cell activation [46]. Moreover, it operates as an immune co-stimulatory factor and has previously been associated with establishing placental immune tolerance during the early stages of pregnancy [47]. However, our investigation revealed elevated levels of sB7-H4 in the serum of PE patients. This observation leads us to speculate that sB7-H4 might exert roles beyond immune tolerance and could potentially play additional adverse roles in the pathogenesis of preeclampsia. In our earlier research, we stated that elevated sB7-H4 levels in serum may be caused by increased B7-H4 expression in placental villi, boosting the Th1-type response [26,27]. As is well known, inflammation, endothelial dysfunction, and poor placentation are all directly associated to Th1-predominant immunity [48]. Furthermore, our recent findings revealed that B7-H4 can directly inhibit the proliferation and migration of trophoblasts, while inducing apoptosis in SGHPL-5 extravillous trophoblast cells [34].

In our study, the primary steps of TPE involved the replacement of plasma components in PE patients with donor fresh frozen plasma or albumin, aiming to reduce anti-angiogenic and inflammatory factors in the plasma and thereby extend the number of gestational weeks in the maternal body [29]. Our findings confirm that TPE treatment indeed leads to a reduction in the serum levels of sFlt-1, sFlt-1/PlGF, sEng, and sB7-H4 in patients. However, when compared to healthy pregnant individuals, these levels remain elevated in PE patients who receive TPE treatment and cannot be dropped down to control levels. Moreover, in our investigation of the corresponding membrane proteins in placental tissues, we observed that, in comparison to preterm controls and PE cases treated with TPE, placental tissues in the PE group without TPE displayed significantly higher levels of Flt-1 and B7-H4. Therefore, we speculate that while TPE treatment can temporarily reduce the circulating levels of sFlt-1 and sB7-H4, the underlying pathophysiological mechanisms of PE persist, leading to the rapid alternative splicing of a substantial amount of Flt-1 and B7-H4 in placental tissues, resulting in the formation of sFlt-1 and sB7-H4, which are subsequently released into the bloodstream.

Although there were no differences in the expression levels of the Eng protein in the placentas of severe PE patients, both in those treated and untreated with TPE, immunofluorescence results revealed a strong detection of Eng in the STB, both apical and basal, in the untreated PE group. When compared to the untreated PE group, the control group and the TPE-treated PE group displayed higher levels of Eng staining in the apical region of STB and less in the basal region, suggesting a treatment effect on the spatial distribution of Eng within placental tissues. Hence, we hypothesize that after TPE treatment, a substantial amount of Eng in the placenta is rapidly converted into sEng and released into the maternal bloodstream leading to the reduction in Eng in placental tissue.

In an inflammatory or pro-inflammatory state, iNOS is activated, and this results in a brief excess of NO. Previous research has demonstrated that NO is a strong inducer of endoplasmic reticulum (ER) stress, which causes chondrocytes to undergo apoptosis [49,50]. Placental tissues from both healthy and hypertensive pregnancies exhibit iNOS expression, which is localized within the villous stroma and extravillous trophoblasts [51,52]. Based on our findings, TPE treatment effectively decreased the expression of iNOS, a protein found to be elevated in PE patients, aligning with previous research [53]. The downregulation of TNF-α and cleaved caspase 3 was also observed in the TPE-treated PE patients compared to the conventionally treated PE patients.

## 4. Materials and Methods

### 4.1. Study Population

A retrospective study of patients diagnosed with severe early-onset PE syndrome was performed at the Department of Gynecology and Obstetrics, University Hospital Essen, Germany, from 2014 to 2019. We included 12 patients with singleton pregnancies and early-onset PE diagnosed before 28 weeks’ gestation who received TPE and compared them with 8 gestational age-matched controls with preterm labor and 12 early-onset PE patients who did not undergo TPE. Standard-of-care treatment was given to all PE patients: a single course of corticosteroid with 12 mg of betamethasone i.m. twice 24 h apart (for acceleration of lung maturation), antihypertensive drugs: Methyldopa maximum 2 g daily and depending on blood pressure, Metoprolol and Nifedipin on top, and magnesium-sulfate i.v. 1 g/h after a loading dose of 4 g. Patients with PE who underwent TPE provided informed consent after a detailed explanation of the procedure before initiating TPE treatment as an experimental therapy option. TPE patients also received standard-of-care treatment.

For blood and placenta analysis, patients enrolled in this study provided signed informed consent. The ethics committee approved the study at the University of Duisburg-Essen, University Hospital Essen, Germany, under the number 12-5212-BO.

Complications unrelated to PE led to preterm birth in the control group due to the following reasons: imminent uterine rupture, glioblastoma, amniotic infection with premature rupture of membranes, intrauterine fetal death, amniotic sac prolapse with premature labor, and cardiotocography pathology.

PE diagnosis adhered to the International Society for the Study of Hypertension in Pregnancy (ISSHP) Guidelines [54], ascertained by the presence of de novo hypertension (systolic >140 mmHg or diastolic >90 mmHg) on at least two occasions measured 4 h apart in previously normotensive women after 20 weeks of gestation. This diagnosis was further characterized by signs of maternal organ dysfunction, involving at least one of the following criteria: proteinuria (>300 mg/24 h), acute kidney injury (creatinine ≥90 umol/L or 1 mg/dL or oliguria), liver disease (elevated transaminases or severe right upper quadrant or epigastric pain), neurological problems, hematological disturbances (thrombocytopenia, disseminated intravascular coagulation, hemolysis), and utero-placental dysfunction (FGR and/or abnormal fetal Doppler sonographic parameters). FGR, also known as intrauterine growth restriction (IUGR), was deemed to be in accordance with the current ISUOG Practice Guidelines [55]. Birth weight percentiles were calculated according to the Fenton growth chart for preterm infants [56].

### 4.2. TPE Treatment

If informed consent was provided following a detailed explanation of the technique, TPE was initiated. TPE was conducted using the Spectra Optia centrifuge system from Terumo BCT, Inc. (Lakewood, CO, USA) and the COM.TEC system from Fresenius (Bad Homburg, Germany). Our primary TPE protocol involved substituting the PE patient’s plasma with fresh frozen plasma (FFP) and/or 4% human albumin. Plasma volume was estimated using the Kaplan formula, with a maximum replacement volume not exceeding 4 L per plasma exchange session. This process was supplemented with anticoagulation, maintenance of acid-base and electrolyte balance, and other relevant therapeutic measures. For the specific operation process, please refer to Iannaccone et al. in 2023 [29]. According to the different conditions of the patients, each patient underwent 1 to 11 TPE treatments; each TPE treatment lasted 2–3 h.

Laboratory testing encompassed routine assessment of renal function, electrolytes, total blood count, transaminases, bilirubin and coagulation. This monitoring was conducted at a minimum frequency of every 12 h during the initial 24 to 48 h, followed by a minimum of three times weekly thereafter. Blood samples were collected at admission, before and within 12 h of every TPE therapy which were used also for sFlt-1, PlGF, sEng and sB7-H4 measurement.

### 4.3. Human Placenta Sample Collection and Processing

Between 2014 and 2019, placental tissues were collected from the Department of Gynecology and Obstetrics at the University Hospital Essen in Germany. Placental tissue was retrieved directly after vaginal birth or caesarean section. This study enlisted the participation of 36 premature pregnant women. The following pathology research groups underwent investigation: 1. control group (N = 8): gestational preterm labor control group without PE or FGR 22 + 4–31 + 6 weeks); 2. PE group without TPE treatment but standard of care treatment (N = 12): pregnant women with very preterm PE without TPE (23 + 5–30 weeks); 3. PE group with TPE treatment (N = 12): early-onset PE pregnant women who received TPE (23 + 2–28 + 5 weeks).

Placental chorionic villous tissue, including the decidua, was cut from the maternal side of the placenta between the umbilical cord and the outer border of the placenta in preparation for immunofluorescence. Only placental chorionic tissue without decidua was collected for RNA and protein isolation. Prior to the extraction of protein samples, tissues were kept at −80 °C.

### 4.4. Enzyme-Linked Immunosorbent Assay (ELISA)

We used the ELISA Kit for sFlt-1 (BRAHMS sFlt-1 KRYPTOR assay, Cat. No. 845.075), PlGF-Plus (BRAHMS PlGF plus KRYPTOR assay, Cat. No. 859.075), human Endoglin/CD105 (R&D Systems, Minneapolis, MN, USA, Cat. No. DNDG00) and V-Set Domain-Containing T-Cell Activation Inhibitor 1 (VTCN1, sB7-H4) (Cloud-Clone Corp., Cat. No. L211008816) to analyze the expression of sFlt-1, PlGF, sEng and sB7-H4 in maternal serum according to the instruction manual. The ELISA reader (TECAN, Model Sunrise; Austria GmbH, Grodig, Austria) and the data analysis software Magellan^TM^ Standard (TECAN, Mannedorf, Switzerland) were used to measure sFlt-1, PlGF, sB7-H4 and sEng levels by absorbance at 450 and 620 nm. The detection limits, intra- and inter-assay % CV were summarized in Appendix A.

### 4.5. Immunoblotting

Preparation of protein extracts followed the protocol reported previously [57]. On a 4–20% polyacrylamide gel (Amersham Biosciences, Piscataway, NJ, USA, Cat. No. 4561094), 20 µg of protein was separated. Next, proteins were transferred onto PVDF membranes. After blocking unspecific binding sites with 5% non-fat milk for 1 h at room temperature, the membrane was incubated with primary antibodies overnight at 4 °C. The primary antibodies listed in Appendix A were employed. At room temperature, the secondary antibody was incubated for 1 h. Detection was performed using the Super Signal West Dura Extended Duration Substrate Kit (Thermo Fisher Scientific, Carlsbad, CA, USA, Cat. No. 34076) and the Chemidoc XRS+ imaging system (BioRad, Feldkirchen, Germany) according to the protocol. Image J2 x (Rawak Software Inc., Wuppertal, Germany) was used to perform a densitometric analysis of each protein bands, after which the protein expression levels were normalized to actin expression. For normalization purposes, the signal value of each blot was normalized to the same “internal control” sample.

### 4.6. Immunofluorescence (IF) Staining

Deparaffinized tissue slices were used, which was followed by antigen retrieval in a citrate buffer at 100 °C for 30 min and permeabilization in 0.3% Triton X-100 in 1× Dulbecco’s Phosphate-Buffered Saline (DPBS) for 20 min. Autofluorescence reducing reagent kit (Max Vision Biosciences, Washington, USA, Cat. No. MB-M) was used to prevent autofluorescence for 5 min, and non-specific sites were blocked by incubating in 0.5% BSA in 1× DPBS for 20 min. Appendix A includes the primary and secondary antibodies used. The primary antibody was incubated for 24 h at 4 °C, followed by a one-hour incubation with the appropriate secondary antibody at room temperature. All samples were counterstained for 15 min at room temperature with 4′, 6-diamidin-2-phenylindol dihydrochloride (DAPI, 1 g/mL, Sigma Aldrich, St. Louis, MO, USA, Cat. No. 28718-90-3) for DNA staining. The primary antibody was omitted for the negative controls. Fluoromount-g mounting medium (Thermo Fisher Scientific, Waltham, CA, USA, Cat. No. 00-4958-02) was used to cover slides before being viewed with a Leica SP5 confocal fluorescence microscope. For each condition, at least three samples of placental tissue were examined.

### 4.7. Statistical Analysis

The sample size was determined by G*Power 3.1 (Heinrich-Heine-University Düsseldorf, Düsseldorf, Germany). GraphPad Prism 9.2 (GraphPad Software Inc., San Diego, CA, USA) was used to perform the statistics. The Grubbs test was used to identify outliers. For comparisons between two groups, the independent samples *t*-test was used if the data followed a normal distribution, whereas the Mann–Whitney U test was applied for non-normally distributed data. For parametric multiple group comparisons, Ordinary one-way ANOVA was employed, while the Kruskal–Wallis test was used for non-parametric multiple group comparisons. The data were presented as either median (interquartile range), minimum and maximum values (range), or mean ± standard deviation. A probability value (*p*-value) of 0.05 or less was required for all statistical tests and was denoted by the symbols * *p* < 0.05, ** *p* < 0.01 and *** *p* < 0.001.

## 5. Conclusions

TPE effectively reduces the levels of sFlt-1, sEng, and sB7-H4 in the maternal circulation of PE patients and prolongs gestational duration. During plasma exchange, membrane-bound proteins (Flt-1, Eng, and B7-H4) within placental villous tissue undergo alternative splicing, resulting in the formation of secreted proteins released into the maternal bloodstream. Thus, we assume that TPE diminished the levels of sFlt-1, sEng, and sB7-H4 in the maternal serum due to a reduced expression of membrane-bound proteins in placental villous tissue. This phenomenon ultimately leads to reduced levels of placental inflammation (iNOS and TNF-α) and apoptosis (cleaved caspase 3) upon TPE.

However, due to the persistent pathophysiological mechanisms underlying PE, these secretory factors cannot reach levels equivalent to those in normal pregnancy.

In conclusion, utilizing TPE to eliminate soluble molecules from the patient’s serum during pregnancy has shown promise in extending gestational duration among early-onset PE patients. This process reduces the soluble molecules (sFlt-1, sEng, and sB7-H4) along with diminishing the expression levels of their placental membrane-bound proteins (Flt-1, Eng, and B7-H4), potentially contributing to improved placental function by lowering inflammation and apoptosis processes in trophoblasts (Figure 6).

TPE represents a promising therapeutic strategy for managing early-onset preeclampsia by targeting and reducing key soluble factors that contribute to the disease’s pathology. Although the secretory factors do not reach levels equivalent to those in normal pregnancies, the significant extension in gestational duration and the reduction in inflammation and apoptosis in placental tissues are noteworthy benefits. Future research should focus on optimizing TPE protocols, understanding the long-term outcomes for both mothers and infants, and exploring the potential integration of TPE with other therapeutic interventions to enhance its efficacy and safety in clinical practice.

### Limitations of This Study

Patients receiving TPE were allocated clinically and suffered from extremely severe early PE. A bias in patient collection is highly probable since the PE standard of care treated group is a retrospectively historically patient cohort with a mean higher gestational age. Consecutively, following prospective studies including exactly matched pairs of patients with and without TPE may be able to show more clearly effects on gestational age at birth or prolongation time gained by TPE. Additionally, the control group does not consist of normal pregnancies, it is a preterm labor group without placental dysfunction and hypertensive disorders but other pathologies like infection or preterm rupture of membranes, the reason for preterm birth. The pathological causes of preterm labor may play substantial roles concerning inflammation parameters. Therefore, the results of this study regarding inflammation markers within the “control” group should be considered with caution.

## Figures and Tables

**Figure 1 ijms-25-11082-f001:**
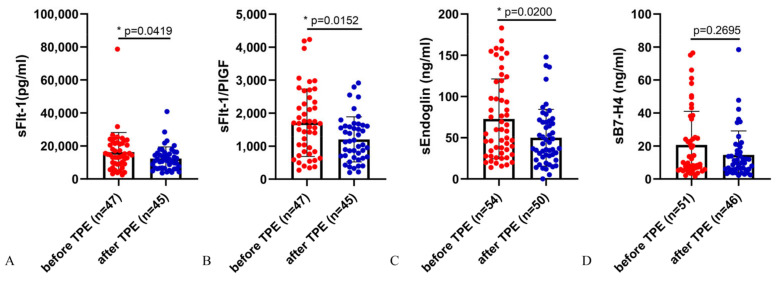
Serum levels of key biomarkers which decreased following therapeutic plasma exchange in early-onset PE patients. Serum levels of sFlt-1 (**A**), sFlt-1/PlGF (**B**), sEng (**C**), and sB7-H4 (**D**) before and after all the TPE treatment in PE patients. Data represent means ± SD. Significance: * *p* < 0.05.

**Figure 2 ijms-25-11082-f002:**
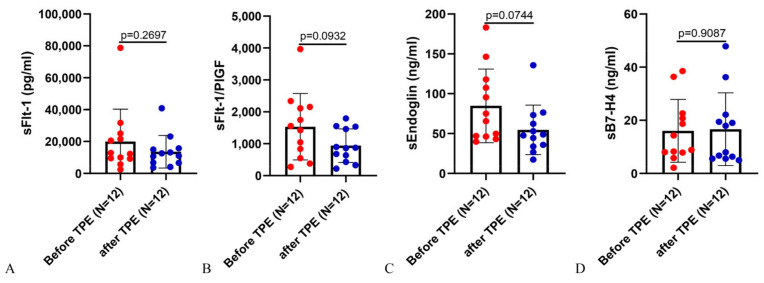
Serum levels of sFlt-1, sFlt-1/PlGF, and sEng, showing a decreased trend but not statistically significant before and after the first therapeutic plasma exchange in PE patients. Serum levels of sFlt-1 (**A**), sFlt-1/PlGF (**B**), sEng (**C**), and sB7-H4 (**D**) before and after the first TPE treatment in PE patients. Data represent means ± SD.

**Figure 3 ijms-25-11082-f003:**
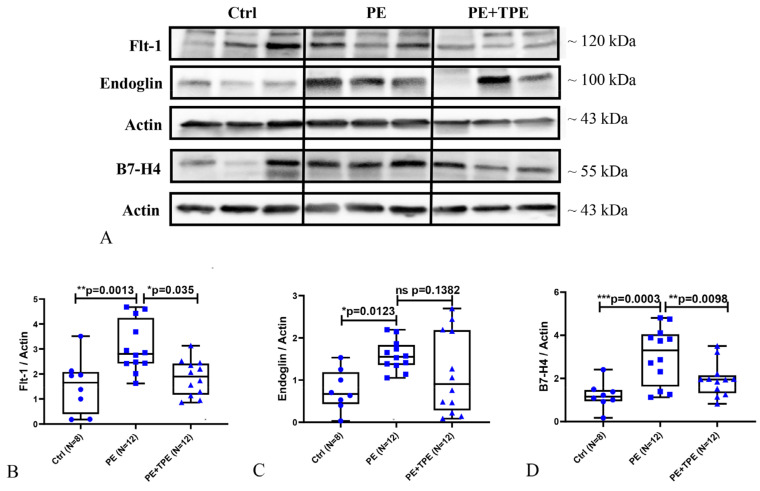
Placental Flt-1, Eng, and B7-H4 protein levels were notably reduced in PE patients treated with TPE compared to those receiving the standard of care treatment. (**A**) Representative immunoblot images of Flt-1, Eng, and B7-H4 protein expression in the placenta of control (N = 8), PE (N = 12), and PE + TPE (N = 12) patients. The original blots are presented in Appendix A. (**B**) Statistical analysis of A to represent the Flt-1 protein levels in placenta. (**C**) Statistical analysis of A to represent the Eng protein levels in placenta. (**D**) Statistical analysis of A to represent the B7-H4 protein levels in placenta. Data represent medians ± interquartile ranges with minimum/maximum values as whiskers. Significance: * *p* < 0.05, ** *p* < 0.01, *** *p* < 0.01.

**Figure 4 ijms-25-11082-f004:**
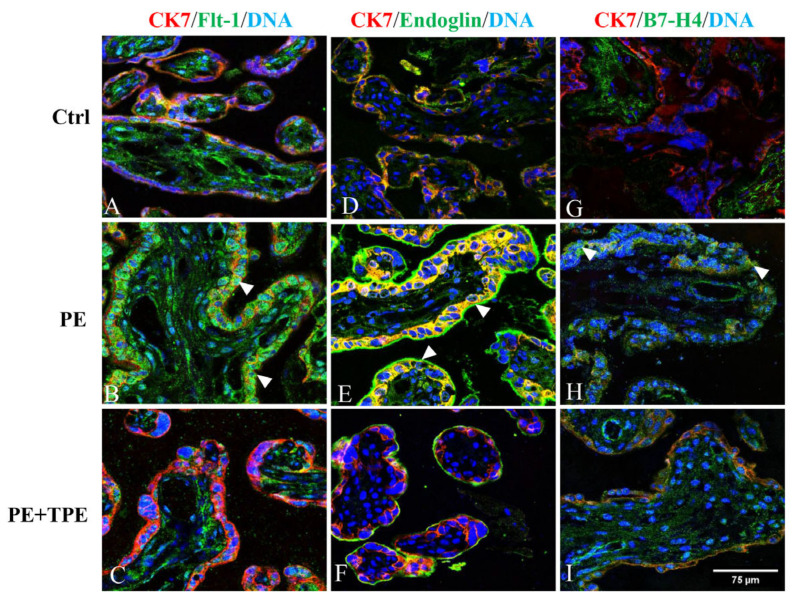
Placental Flt-1, Eng, and B7-H4 protein levels seem to be decreased in placental villi in PE patients treated with TPE compared to those receiving the standard-of-care treatment. (**A**–**C**) Double immunolabeling of Flt-1 (green) and CK7 (red) in placentas from controls and PE patients without and with TPE treatment. The STBs show strong Flt-1 staining in the untreated PE group. (**D**–**F**) Double immunolabeling of Eng (green) and CK7 (red), highlighting reduced Eng staining on the basal side of the STB in TPE-treated patients. (**G**–**I**) Double immunolabeling of B7-H4 (green) and CK7 (red), with the untreated PE group showing higher B7-H4 expression in the STB. DAPI was used to counterstain DNA (blue). The triangles point to the STB. The scale bar represents 75 μm.

**Figure 5 ijms-25-11082-f005:**
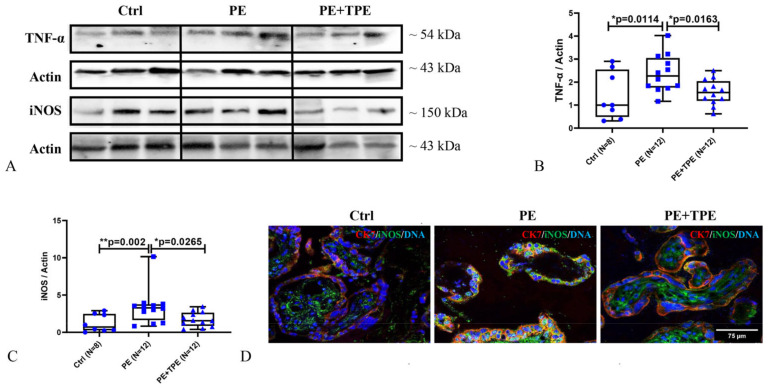
Placental inflammation was notably reduced in PE patients treated with TPE compared to those receiving the standard-of-care treatment. (**A**) Representative immunoblot images of TNF-α and iNOS protein expression in the placenta of control (N = 8), PE (N = 12), and PE + TPE (N = 12) patients. (**B**,**C**) Statistical analysis of A. (**D**) Double immunolabeling of iNOS (green) and CK7 (red) in placentas of control, PE, and PE + TPE patients. DAPI was used to counterstain DNA (blue). Data represent medians ± interquartile ranges with minimum/maximum values as whiskers. Significance: * *p* < 0.05, ** *p* < 0.01. The scale bar represents 75 μm.

**Figure 6 ijms-25-11082-f006:**
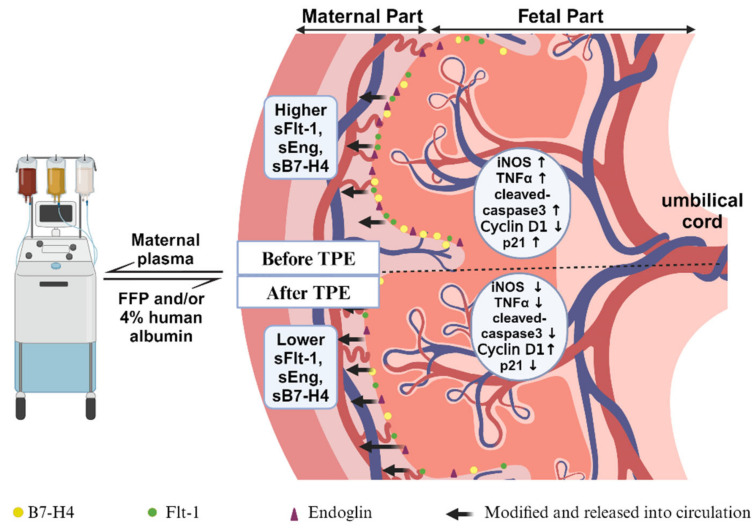
Schematic diagram of TPE treatment in early-onset PE. TPE demonstrated efficacy in extending the gestational period for patients with early-onset PE. TPE removes a portion of maternal plasma, replacing it with fresh frozen plasma and/or 4% human albumin, leading to a reduction in these circulating factors. During plasma exchange, there is a notable reduction in the circulating maternal levels of sFlt-1, sEng, and sB7-H4, alongside a corresponding decrease in the membrane-bound proteins (Flt-1, Eng, and B7-H4) within placental tissues, which are the original source of these secreted proteins. This reduction aligns with an amelioration in placental functionality. Specifically, the observed decrease in placental inflammation (iNOS and TNF-α) and cellular apoptosis (cleaved caspase 3), along with the increase in proliferation (Cyclin D1, p21) after TPE treatment, may potentially be associated with the role of B7-H4. Diagram created with BioRender.com.

**Table 1 ijms-25-11082-t001:** Clinical characteristics of pregnant women and newborns in control group, PE patients receiving TPE treatment versus PE patients receiving standard-of-care treatment.

Variable	Control	PE	PE + TPE	*p* Value
Maternal age at delivery, years, mean ± SD	32.13 ± 6.56	32.33 ± 4.74	30.92 ± 6.24	0.8190
Gestational age at delivery, weeks, mean (min and max)	28 (22 + 4–31 + 6)	27 + 1 (23 + 5–30)	25 + 3 (23 + 2–28 + 5)	0.1010
Pregnancy BMI before birth, mean ± SD	31.71 ± 7.22	30.30 ± 7.57	33.81 ± 8.65	0.5097
Cesarean section, no. (%)	75	100	100	-
Systolic blood pressure, mmHg, mean ± SD	123.50 ± 9.87	153.10 ± 25.79	160.50 ± 15.69	<0.001 ^†#^
Diastolic blood pressure, mmHg, mean ± SD	72.00 ± 11.29	89.58 ± 18.73	88.67 ± 11.69	<0.05 ^†#^
Proteinuria, mg/24 h, mean ± SD	-	1806 ± 2180	2908 ± 1888	0.1754
Platelet count, cells/mm^3^, mean ± SD	281.5 ± 70.77	165.0 ± 79.42	179.8 ± 67.57	<0.05 ^†#^
AST (GOT), IU/L, mean ± SD	23.00 ± 8.74	36.50 ± 19.58	20.83 ± 8.91	0.0591 ^§^
ALT (GPT), IU/L, mean ± SD	25.83 ± 17.38	43.75 ± 35.55	19.75 ± 11.22	0.3703
LDH, IU/L, mean ± SD	216.8 ± 50.88	287.1 ± 91.35	222.4 ± 74.32	0.2257
Creatinine, mg/dL, mean ± SD	0.60 ± 0.09	0.89 ± 0.50	0.69 ± 0.14	<0.05 ^†^
Birth weight, g, mean ± SD	1104 ± 544.2	970 ± 781.3	552.7 ± 258.7	<0.05 ^#^
Birth weight percentile, mean ± SD	35.00 ± 22.64	12.25 ± 8.18	8.96 ±10.11	<0.05 ^†#^
5 min Apgar score, mean ± SD	7.60 ± 1.14	7.92 ± 1.08	7.46 ± 1.70	0.8362
Neonatal survival rate, no. (%)	75.00	83.33	58.33	-
sFlt-1 at delivery, pg/mL, mean ± SD	3547 ± 2759	18,336 ± 9569	16,788 ± 10,326	<0.05 ^†#^
sFlt-1/PlGF at delivery, mean ± SD	89.41 ± 130.4	1789 ± 1712	1550 ± 908.2	<0.05 ^†#^
sEndoglin at delivery, ng/mL, mean ± SD	-	88.26 ± 35.52	62.80 ± 49.44	0.2800
sB7-H4 at delivery, ng/mL, mean ± SD	-	20.10 ± 18.93	12.02 ± 9.99	0.8317
Pregnancy prolongation, day, median	-	3.5	6.5	0.1716

Abbreviations: ALT(GPT), alanine aminotransferase; AST(GOT), aspartate aminotransferase; LDH, lactate dehydrogenase; nm, not measured. † denotes *p* < 0.05 between the control and non-TPE-treated group; # means *p* < 0.05 between the control and TPE-treated group; § means *p* < 0.05 between the non-TPE-treated and TPE-treated group.

**Table 2 ijms-25-11082-t002:** sFlt-1, sFlt-1/PlGF, sEng, and sB7-H4 levels measured before and after TPE in the PE + TPE patient group.

		Before TPE	Delta%	After TPE	*p* Value
sFlt-1 (pg/mL)	Median	14,590	−24.81%	10,970	0.0419
	Mean	16,323	−24.22%	12,370	
	SD	11,817	−39.84%	7109	
sFlt-1/PlGF	Median	1685	−32.94%	1130	0.0152
	Mean	1713	−27.61%	1240	
	SD	1023	−33.18%	683.6	
sEndoglin (ng/mL)	Median	58.89	−32.25%	39.90	0.0200
	Mean	72.82	−31.28%	50.04	
	SD	48.45	−28.98%	34.41	
sB7-H4 (ng/mL)	Median	10.02	−0.70%	9.95	0.2695
	Mean	20.68	−29.40%	14.6	
	SD	20.38	−28.66%	14.54	

## Data Availability

The datasets generated and/or analyzed during the current study are available from the corresponding author.

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
