# Peer review of "Alteration in sB7-H4 Serum Levels and Placental Biomarker Expression after Therapeutic Plasma Exchange in Early-Onset Preeclampsia Patients"

_ijms, 2024, doi:10.3390/ijms252011082_

Round 1

Reviewer 1 Report (New Reviewer)

Comments and Suggestions for Authors

Studying pre-eclampsia is challenging and devising an experimental therapy based on molecular medicine is impactful and important. I would like to commend the study team for their great efforts.

I do some questions which I hope the authors can address:

1. I note that there is power calculation for the comparison cohorts, be they historical; will the aetiology of the pre-term labour impact directly on the biomarkers studied? 

2. Will the use of anti-hypertensives and the choices or even the use of Magnesium sulphate impact on the vessel remodelling and biomarkers measured in the study? There is some evidence e.g. PMC7664615, that this may result in the findings observed and it was only mentioned standard of care - I would like to find out if the TPE group also received "standard of care"? It would be important to factor that in the interpretation of the condition. Also, for the pre-term labour cases, is magnesium sulphate used for neuroprotection for early delivery based on the aetiologies discussed? 

3. It would be ideal for the authors to propose on how TPE should be employed in the management of PE other than the standard of care or should this be reserved for the early preterm PE?

Comments on the Quality of English Language

English language is generally fine.

Author Response

Reviewer 2 Report (New Reviewer)

Comments and Suggestions for Authors

Duan et al. present a well-structured manuscript, with a conservative but adequate methodology. This is a manuscript that addresses a topic of interest and within the scope of this Journal. The results are correct and allow for solid conclusions. However, the authors should address indications to improve the final quality of the manuscript.

-The authors should address an improvement of the summary of the manuscript. The authors should include a more clinical and translational conclusion. This point should be necessary to attract the reader's attention.

-The authors should include more clinical terms in the key words.

-The authors should include in the introduction the pathophysiological justification of the mechanisms involved in vascular pathology as a whole.

-The authors should improve the presentation of the results. The graphs are of very low quality. The graphs should be larger and much better described.

-The figures legends are very brief, the authors should improve them and put data in a more precise way.

-The authors should make a better description of the fluorescence figures. The description of the placental structures is very limited. It should be marked both in the text of the manuscript and in the figures.

-The graphical summary is adequate, but should be better described.

-The authors make an adequate discussion, but the clinical data are not taken into account accurately. The connection should be greater to give greater solidity to the manuscript.

-The methodology is correct. Please, the authors should improve the justification of the statistical methodology and the sample size.

-The authors should correct the small style errors accurately.

Comments on the Quality of English Language

Minor editing of English language required.

Author Response

This manuscript is a resubmission of an earlier submission. The following is a list of the peer review reports and author responses from that submission.

Round 1

Reviewer 1 Report

Comments and Suggestions for Authors

In this manuscript authors investigated the impact of TPE on serum sB7-H4 and placental proteins such as Flt-1, Eng, B7-H4, iNOS and TNF-α in TPE-treated early-onset PE patients, conventionally treated counterparts and gestational age-matched controls. Authors found that  TPE extended pregnancy by a median of 6.5 days in this cohort of early-onset PE. Serum sB7-H4, sFlt-1, and sEndoglin levels decreased, along with reduced expression of their membrane-bound proteins in placental tissue upon treatment with TPE. Moreover, TPE-treated patients also showed a decreased placental inflammation compared to the non-TPE group.

The manuscript is interesting, generally well written and well illustrated. However, there are several points that deserves to be improved. See comments below.

Line 6: remove "and" at the end of authors' list

Line 11: Corresponding author is missing

Line 27: remove bold 

Lines 31-49: An introduction of PE pathophysiology, classification (especially early- and late- onset)  is necessary.  Moreover, PE risk factors such as high BMI, previous PE pregnancy, advanced age, and paternal factors (see PMID: 31536940 and PMID: 38534435) deserves to be mentioned since are often ignored. 

Line 50-52: it deserves to be highlighted that PE pregnancies are also characterized by an increased oxidative stress (see PMID: 37296665 ) due to the placental hypoxic environment characterizing this pathology. 

Lines 98-99: References must be added

2.4. Enzyme-Linked Immunosorbent Assay (ELISA): I suggest to move the ELISA kits used in a dedicate table. In the same table, detection limits and intra- and inter-assay % CV should be added. 

Line 196: Sudan Black concentration and incubation time must be added

Table 1: Gestational age at delivery should be reported in weeks 

Figure 4 and 5D: Higher magnifications are needed

Limitations of this Study: Authors must add an important limitation of the study. In fact, it must be pointed out that the gestational ages were significantly lower in PE+TPE group compared to the control group. This is a very important limitation since several of the values reported in table 1 significantly change with the progress of gestation. 

Authors must add the product codes of all reagents used

Authors must write the abbreviations in full length when mentioned for the first time

Reviewer 2 Report

Comments and Suggestions for Authors

Check the English language! For example, line 117-118 is incorrect: "Ana all methods were performed in accordance with the relevant guidelines and regulations." I have noticed several ambiguous expressions where I believe they intended to convey something else.

The preeclampsia groups were compared with preterm pregnancies. The advantage of this study is its rarity in the literature. However, this is also its drawback, as there is no healthy control group. Comparing with healthy controls would be more beneficial for understanding preeclampsia. Blood samples taken at the same gestational age could serve as a good healthy control sample. IUGR is determined by gestational age, which could also be utilized.

Comments on the Quality of English Language
  1. sFlt-1, sEng, and sB7-H4 are generated not through limited proteolysis but through alternative splicing! Please revise this wording throughout the text.
  2. Introduction The authors describe the functions of the proteins studied. I recommend including information about sFlt (it reduces the amount of effective VEGF by binding to it. Its excessive expression in preeclampsia can result in insufficient angiogenesis).
  3. 2.1. Study Population The chapter introduces the ISSHP definition of preeclampsia, but only patients with severe preeclampsia were included in the study. Please add the definition of severe preeclampsia to the text! The control group was a corrected (preterm) group based on the delivery time. Therefore, it is justified to present the clinical data of the control group as well! Pay particular attention to data that increase the risk of preeclampsia (e.g., pregnancy hypertension, GDM, kidney disease, BMI > 30, etc.).
  4. 2.2. TPE Treatment What was the indication for TPE treatment? Was knowledge of the study and consent for treatment sufficient?
  5. 2.3. Human Placenta Sample Collection and Processing Describe in the methodology how placental RNA was protected from degradation.
  6. 2.5. Immunoblotting Why were the PVDF membranes not blocked?
  7. 2.6. Immunofluorescence (IF) Staining

a) I know what PBS (line 195) is, but please write out what it stands for in the text.

b) Why was DAPI used? Include in the text that it is suitable for DNA staining! DAPI staining is not mentioned in the results (I can see it in Figures 4 and 5).

  1. I suggest correcting Table 1.

a) Gestational age is generally given in weeks, not days. The commonly used values are easier to interpret.

b) Include some additional data:

    • How many (n or %) participants had elevated blood pressure (systolic or diastolic) above the threshold?
    • IUGR
    • Some SD values (sEndoglin, sB7-H4) are missing

a                c) p column

                     - Where two significant differences were found, to which does the value refer?

                    - It would be simpler to use p < 0.05; p < 0.01; p < 0.001 symbols.

  1. Table 2. Include the threshold values as well!

  1. Figures 4 and 5 Describe what the colors represent!

Reviewer 3 Report

Comments and Suggestions for Authors

- why is GA p vlaue significant if groups were matched according to GA?

- overall comparison with p value across all 3 groups makes results unclear

- EFW and BW percentiles? were they PE or severe PE?

- reason for preterm delivery in control group? inflammation excluded?

- follow up of mothers?

- TPE: prolongation of pregnancy but at what cost to mothers and fetuses/neonates?

- correlation with other markers?

- did the patients receive ASS prophylaxis?

Comments on the Quality of English Language

minor revision required as well as grammar and some sentences are not completed (cf. section 2.1)

Round 2

Reviewer 1 Report

Comments and Suggestions for Authors

the manuscript can be accepted in the current form

Author Response

Thank you for your feedback and for approving our revised manuscript.

Reviewer 2 Report

Comments and Suggestions for Authors

The manuscript has been sufficiently improved to warrant publication in IJMS. 

Author Response

(The authors gave the same response as above.)

Reviewer 3 Report

Comments and Suggestions for Authors

- I am still not happy with the gestional age differences. You claim in Abstract and Methods that this is how you were able to compare the groups, yet this is a major limitation. Was this not clear and obvious before you started your study, since cases were more severe?

- depiction in Table 1 is confusing, add lines or more space between parameters

- definition of severe PE does not include gestational age, but you just assume that lower GA was more severe?

- what percentiles were used for birth weight? FMF?

- all the causes you listed for preterm delivery potentially include inflammation!

- A final summary/conclusion and outlook is missing at the very end of the conclusion section

- revision of English language necessary

Comments on the Quality of English Language

- revision of English language necessary
